# YOLO-plum: A high precision and real-time improved algorithm for plum recognition

Yupeng Niu[1,2], Ming Lu[1,2], Xinyun Liang[1,2], Qianqian Wu[1,2], Jiong Mu[1,2]*

1 College of Information Engineering, Sichuan Agricultural University, Ya'an, China, 2 Sichuan Key Laboratory of Agricultural Information Engineering, Ya'an, China

* jmu@sicau.edu.cn

## Abstract

Real-time, rapid, accurate, and non-destructive batch testing of fruit growth state is crucial for improving economic benefits. However, for plums, environmental variability, multi-scale, occlusion, overlapping of leaves or fruits pose significant challenges to accurate and complete labeling using mainstream algorithms like YOLOv5. In this study, we established the first artificial dataset of plums and used deep learning to improve target detection. Our improved YOLOv5 algorithm achieved more accurate and rapid batch identification of immature plums, resulting in improved quality and economic benefits. The YOLOv5-plum algorithm showed 91.65% recognition accuracy for immature plums after our algorithmic improvements. Currently, the YOLOv5-plum algorithm has demonstrated significant advantages in detecting unripe plums and can potentially be applied to other unripe fruits in the future.

## Introduction

To optimize economic benefits in fruit tree farming, it is vital to proactively monitor their condition and take appropriate actions in a timely manner. Through detecting the mature status of fruit trees, farmers can carry out a suite of operations, including intelligent spraying, to improve productivity and resource efficiency. This approach not only maximizes yields but also promotes a sustainable agricultural industry with improved profits.

Plums are one of the three major fruits in China with a history of over 3,000 years and hold a significant share in the Chinese market. However, traditional methods of plum planting, fertilization, and picking require a considerable amount of manpower and resources. Precision planting techniques such as intelligent spraying, growth detection, and intelligent picking can optimize resource allocation, accurately estimate yield, and predict labor demand.

In practice, Chinese farmers usually harvest plums before they ripen to ensure quality, facilitate transportation, storage, and processing. At this stage, the plum skin is typically cyan or light cyan, which provides optimal conditions for accurate detection of unripe plums using computer vision. However, detecting blocked or overlapping fruits in natural scenes remains a challenge, requiring the development of a quick and accurate batch detection method.

**Data Availability Statement:** All relevant data are on Figshare: https://doi.org/10.6084/m9.figshare.23641542.

**Funding:** The authors received no specific funding for this work.

**Competing interests:** The authors have declared that no competing interests exist.

Efforts to improve plum production efficiency require accurate yield estimation and early prediction of labor demand. Innovative technologies, such as intelligent picking, can enhance the efficiency and accuracy of fruit picking. Additionally, the use of intelligent spraying can reduce the use of pesticides, while growth detection can provide timely information on the status of the fruit.

In conclusion, the adoption of precision planting and intelligent technologies can significantly enhance plum production in China, optimizing resource allocation, reducing pesticide use, and improving the efficiency and accuracy of fruit picking. These technologies address the challenges associated with traditional plum production and processing and can improve the quality and yield of plums in the Chinese market.

In recent years, computer vision has been increasingly applied to fruit detection, and many researchers have combined computer vision with fruit recognition and detection. Currently, there has been some progress in the research of fruit target detection in natural scenes. Peng et al. [1] proposed to use shape moment invariants and other methods to integrate fruit color and shape features, and use SVM classifier to classify and recognize the extracted feature vectors. However, this method has poor adaptability in complex environment. Yang Jiangping [2] proposed the method of fruit and vegetable classification by extracting the Gabor features of fruit and vegetable images, then reducing the dimensions of the extracted features, and then using the support vector machine (SVM) method for classification. Experiments show that this method can improve the recognition accuracy effectively. Zhao et al. [3] detect unripe green citrus by combining color characteristics and absolute transformation difference sum, so as to predict the yield and profit of ripe citrus before the harvest period. However, there is also the problem of missing fruit detection caused by leaf shading or uneven illumination. With the development and application of hyperspectral technology, more and more scholars begin to use it for research. For example, Francesca et al. [4] use hyperspectral technology to determine the optimal harvesting period of grapes. Five grapes harvested at different periods were collected, which were randomly selected for a total of 180 spectra in the top, middle and bottom regions. Different spectral pretreatment methods were used to establish independent SIMCA models and PLS-DA models. The accuracy of SIMCA was lower than that of PLS-DA model. Using PLS-DA model, the accuracy of grape identification in the fifth harvest period was 94%, and the accuracy of all classification was 100%. Tao et al. [5] studied a machine vision system that can detect the skin color of apples and potatoes. By using multivariate recognition technology, it can recognize the skin color with 90% accuracy, but its detection efficiency is not high. Hou and Lei et al [6]. proposed a fruit recognition algorithm, which improved the recognition rate by proving that the recognition rate of CNN combined with selection search algorithm was higher than that of traditional CNN alone. To evaluate their model, they used a small data set with images of fruit against a white background, but their model did not examine the overlap of fruit. Ji et al. studied an algorithm for apple classification and recognition based on support vector machine. However, for apples shielded by leaves, there will be missing detection, and the efficiency of apple recognition is low [7]. Sabzi et al. [8] extracted 452 features from 100 visible images of each of the three oranges, trained with mixed ANN-ABC, and established PH estimation model for oranges. Verification set verified that the PH estimation effect of the established model for the three oranges met the accuracy requirements. Xu et al. [9] extract Histogram of Oriented Gradients of strawberry by using HOG (Histogram of Oriented Gradients) to detect strawberry through SVM(Support Vector Machine). The accuracy of this method was 87%, and the average detection time of single and overlapping grass berry was 1s and 2s, respectively. In recent years, object detection based on deep convolutional neural network [10] has been in a leading position in the whole detection field. At present, there are

two classification standards for object detection algorithms. In this paper, only the first type is discussed. The algorithm of generating candidate regions, then classifying and regressing the candidate regions is called two-stage algorithm [11]. Algorithms that do classification and regression without generating candidate regions are called one-stage algorithms [12]. The two-stage algorithm has high accuracy but poor real-time performance, and the effect of detecting small targets is poor. The one-stage algorithm has the advantage of high real-time performance, but the accuracy of group targets and small targets is low, and the suitable scene is real-time target detection [13]. However, in real life, when performing time-sensitive operations such as fertilizer application and fruit picking, the detection of plants is often real-time target detection, and its significance is greater. Therefore, in the detection process of immature plums, this paper uses a stage algorithm in the deep convolutional neural network to carry out subsequent experiments, including classic networks such as SSD [14] and YOLO series [15, 16]. But inevitably, the drawback of the one-stage algorithm is its low accuracy problem for group targets and small targets.

In order to address the challenges of real-time, fast detection [17] and high accuracy [18] in identifying immature plums in natural scenes, this paper utilizes YOLOv5, which is known for its high accuracy and fast detection speed in the field of image detection. To further improve the model, we reference the SPPCSPC improvement idea compared to SPP [19], and modify the convolutional layer structure of the SPPF in YOLOv5. We also change the parallel structure to a more lightweight series structure to reduce the number of parameters in the feature extraction process. Additionally, we change the nearest neighbor interpolation [20] used in the original model to Bi-Cubic interpolation [21] and adopt a multi-scale training approach.

Specifically, our improvements include:

(1) Improving the SPPCSPC structure to obtain the SPPFCPC structure, which increases detection speed while maintaining the same level of accuracy.

(2) Replacing the nearest neighbor interpolation of the YOLOv5 upsampling module with cubic convolution interpolation to improve prediction accuracy.

(3) Incorporating multi-scale training by training on larger and larger images to improve the robustness of the detection model to object size.

As a result, we trained an improved YOLOv5-Plum object detection network that effectively addresses the challenges of real-time, fast detection and high accuracy when identifying immature plums in natural scenes.

The experimental results clearly demonstrate that the proposed YOLOv5-Plum object detection network effectively enhances the speed of YOLOv5 feature extraction without sacrificing accuracy. Moreover, it minimizes resolution loss during the upsampling process, thereby enabling maximum extraction of feature values from the image. This method significantly enhances the network's robustness and improves detection accuracy, ultimately resulting in a marked improvement in the network's detection effectiveness.

In the following sections, we provide a detailed description of the dataset used in this study and an overview of the experimental platform. We also discuss the improvements we made to the network module and network construction, as well as the process of model training and evaluation. Additionally, we present the experimental results, including a comparison of our proposed YOLOv5-plum model with other existing models, as well as an ablation experiment of each algorithm. We further discuss the contributions and limitations of our network, as well as potential directions for future research. Finally, we offer our conclusions and provide further insights for future work in this area in section 5.

## Materials and methods

### Experiment field and data acquisition

The experimental orchard used in this study is situated in Gulin County, Luzhou City (Fig 1), Sichuan Province (see Table 1 for detailed geographical location), and the plum variety examined is the crisp red plum. The Chishui River Basin, where the plum is grown, is one of the four major subtropical fruit producing areas in China, with abundant sunlight, a large temperature difference between day and night, and moderate rainfall. The Chishui River is the only river with natural flow and good authenticity among the first-level tributaries of the upper Yangtze River. Therefore, the area it flows through is a core source of oranges, peaches, and plums. Among these basins, Gulin County is renowned for its "high yield and excellent quality" crisp red plums, which are generally spherical in shape. The immature pericarp is green or green, the pericarp near physiological maturity is red and green, and the pericarp is dark red after maturity.

To ensure sample diversity and improve the robustness of the network, plum images were captured using a variety of devices, including Xiaomi 11 pro (Xiaomi Technology Co., LTD., Beijing, China), and Canon camera (PowerShot SX30 IS, 18-megapixel DIGIC4, 24–840 IS lens). The dataset used in this study was collected from two different devices and includes a variety of scenarios depicting plums under different conditions such as sunny, cloudy, branch-

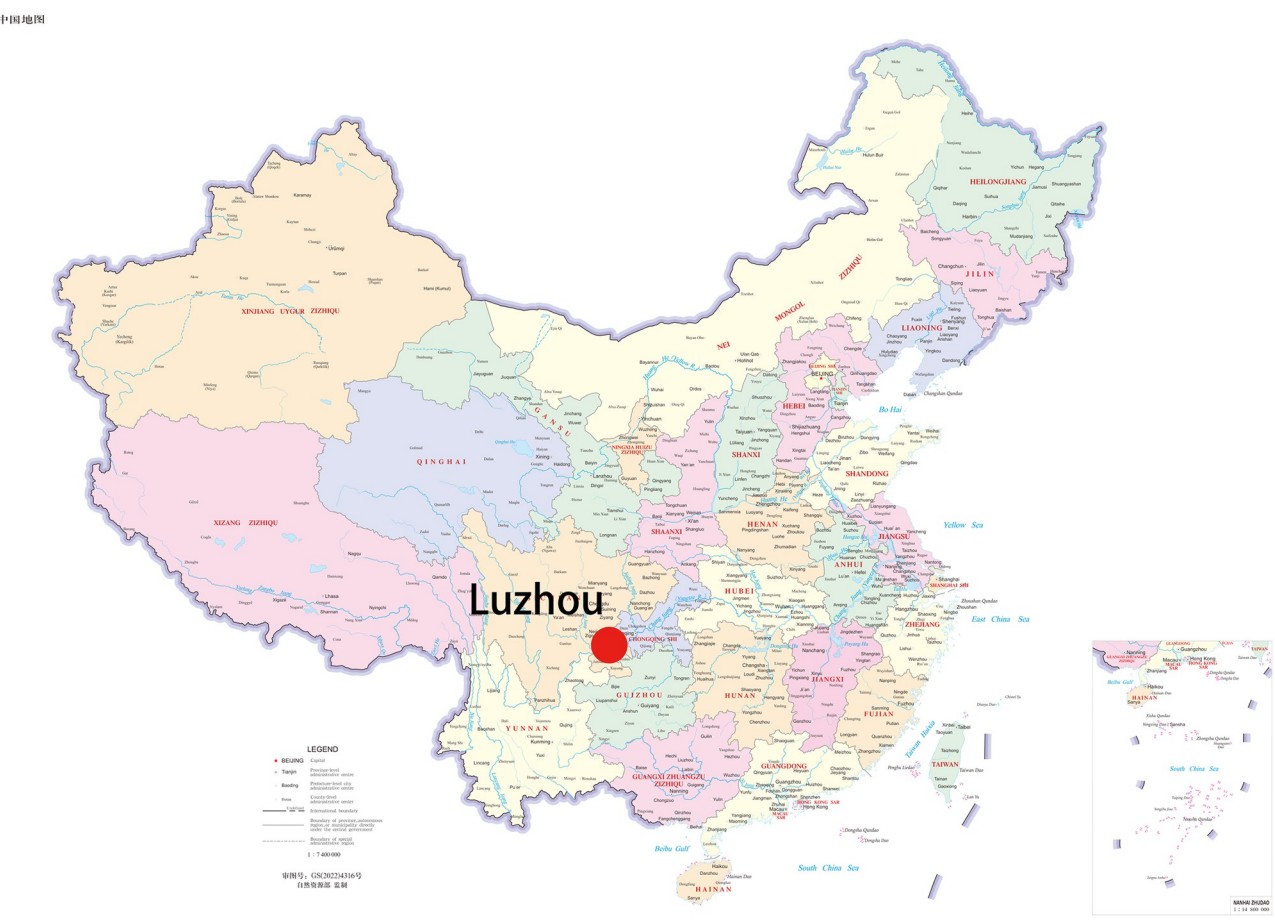

中国地图

Luzhou

**Fig 1. Location of the experimental area in LuZhou, as marked on the map.**

**Table 1. Information about the plum dataset.**

| Place of Collection | Gulin County |
|---|---|
| Country | The People's Republic of China |
| Administrative Region | Luzhou City, Sichuan |
| under Jurisdiction | Province |
| Coordinate | 28°02′39″N 105°48′31″E |
| Altitude (m) | 510–520 |
| Label Name | Plum |
| Plant Variety | Crisp red plum |
| No. of Pictures | 2624 |
| No. of Training Sets | 976 |
| No. of Validation Sets | 243 |

shaded, overlapping fruits, and small targets. The dataset takes into account various factors such as lighting and obstructions, as shown in Fig 2 which displays a selection of images captured by the two devices. In this study, a total of 2624 plum images were initially captured. After a thorough manual screening process to remove duplicate and similar data, a final dataset of 1219 images was obtained, which consisted of 976 images in the training set and 243 images in the validation set.

**Experimental platform.** This research experiment was carried out using state-of-the-art technology and equipment. The frame image source was PyTorch 1.8.1 and the training environment was Python 3.8 (Ubuntu 18.04) with Cuda 11.1 as the computing architecture. The hardware GPU was an RTX 3090*1 with a video memory of 24GB, and the CPU used was a

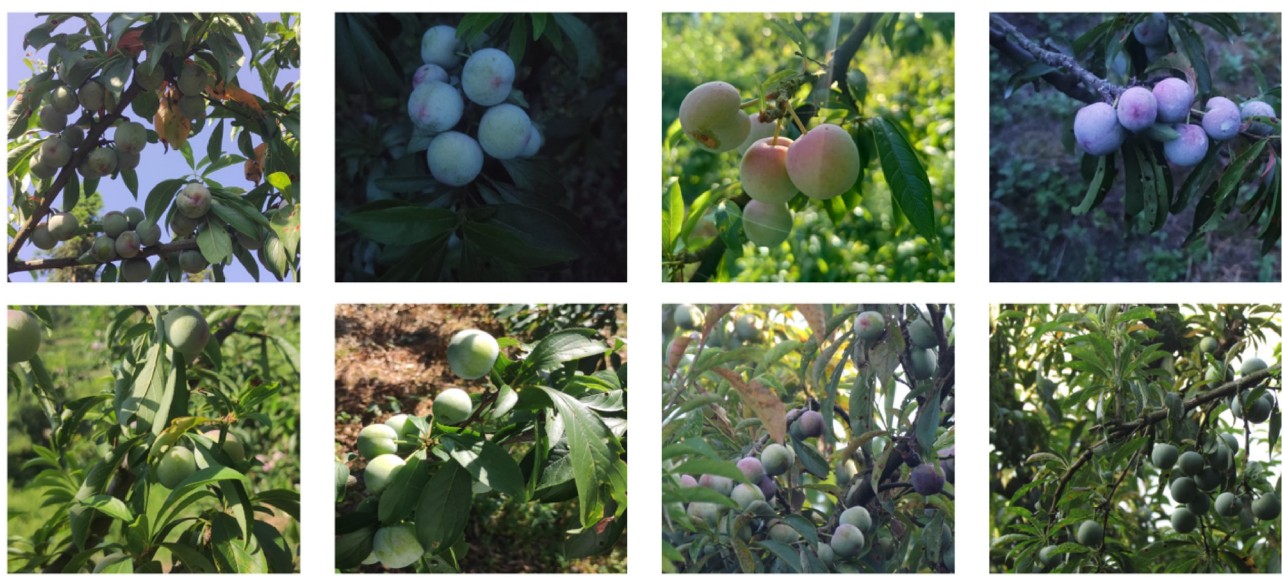

**Xiaomi 11 pro                    CANON PowerShot SX30 IS**

**Fig 2. Examples of images taken by different devices.**

12-core Intel Xeon(R) Platinum 8260C CPU @2.40GHz, which ensured efficient and accurate training of the neural network models used in the study.

## Deep learning network construction

**SPPFCSPC module.** As illustrated in Fig 3, the Spatial Pyramid Pooling (SPP) module was first introduced by He Kiming et al. in 2015 [19]. This module utilizes a spatial pyramid pooling structure [22–26] to effectively address image distortion issues that arise from region clipping and scaling operations. Moreover, it resolves the problem of repeated feature extraction of graph-related convolutional neural networks. By significantly improving the speed of generating candidate boxes, SPP greatly reduces computational costs. In summary, SPP represents a novel approach to pooling in deep neural networks, which can enhance the efficiency and accuracy of object detection algorithms.

As shown in Fig 4, SPPF(Spatial Pyramid Pooling -Fast) is YOLOv5 proposed by Glenn Jocher based on SPP, which replaces a 9*9 convolution operation with 2 5*5 convolution operations, and a 13*13 convolution operation with 3 5*5 convolution operations. In the case of ensuring the same receptive field, the output of each pooling will become the input of the next pooling, which reduces the calculation time and is much faster than SPP.

As shown in Fig 5, SPPCSPC is the SPP structure used in YOLOv7 [27], which makes some changes on the traditional SPP structure, and introduces CSP structure on the basis of the traditional SPP structure. SPPCSPC performs better than SPPF on COCO dataset, but the disadvantage is that the number of parameters and calculation amount are greatly increased.

This paper refers to the improvement idea of SPPF compared with SPP, improves the structure of SPPCSPC, and obtains the SPPFCSPC module, which can also improve the speed while maintaining the same receptive field. The SPPFCSPC structure diagram is shown in Fig 6 below.

**Bi-Cubic interpolation.** Upsampling means that on the basis of the original image pixels, a suitable interpolation algorithm [28–30] is adopted between pixel points to insert

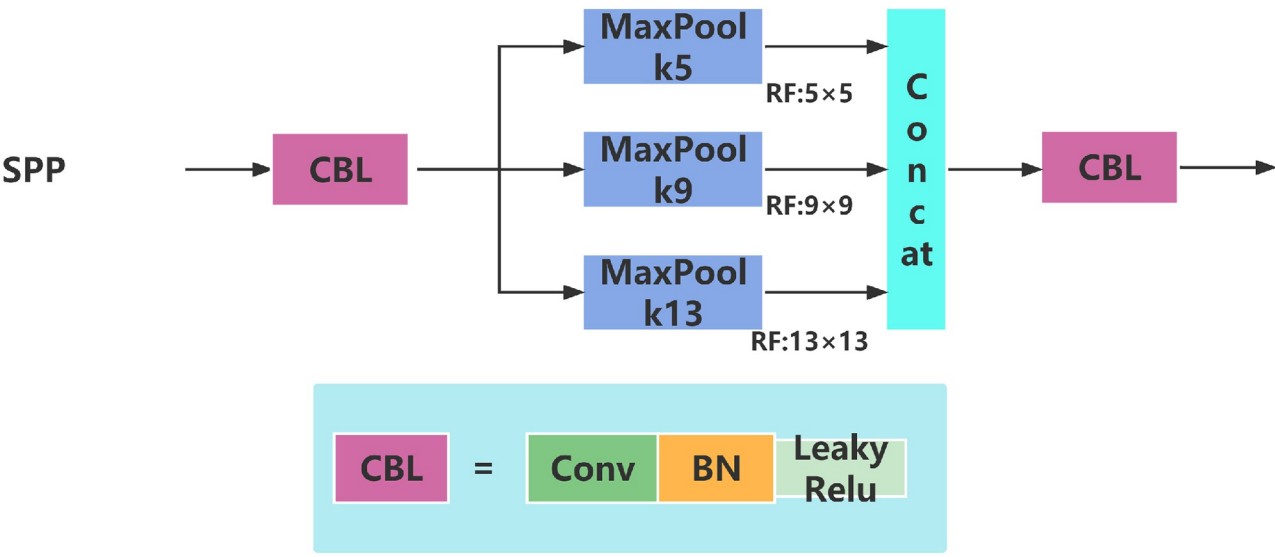

**Fig 3. SPP structure diagram.**

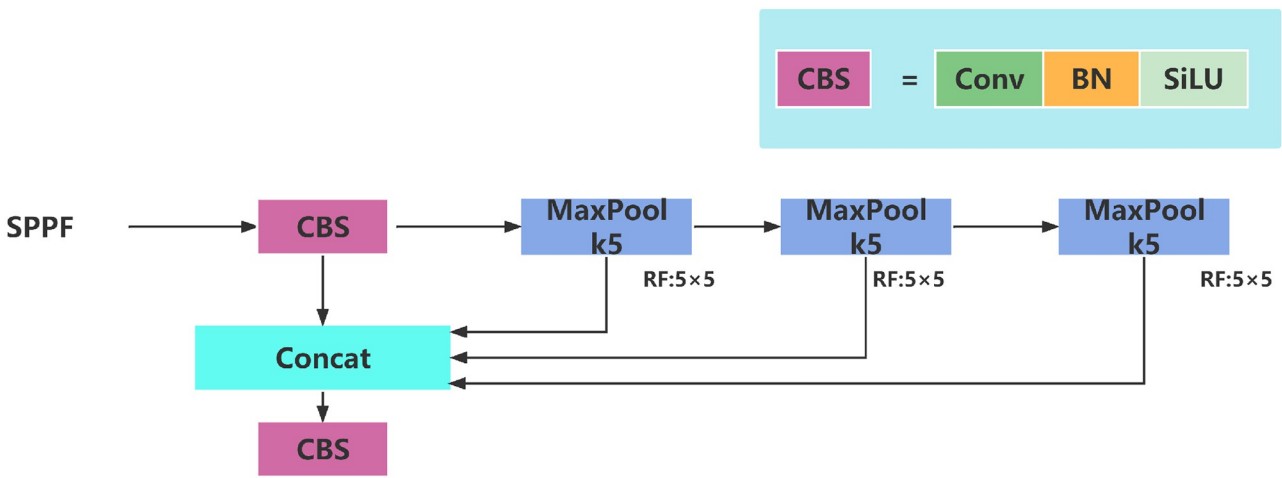

**Fig 4. SPPF structure diagram.**

new elements, that is, to enlarge the original image, so that the image becomes a higher resolution [31]. Common interpolation algorithms include nearest neighbor interpolation, bilinear interpolation, cubic convolution interpolation [28], etc. In simple terms, nearest neighbor interpolation works by finding the nearest pixel to the position of the inserted pixel and inserting it as its pixel value. The bilinear interpolation calculated the pixel value of the inserted pixel position according to the pixel value of the nearest four points. The cubic convolution interpolation obtains the pixel value of the inserted pixel by the weighted average of the 16 nearest pixels in a rectangular grid to the position of the inserted pixel.

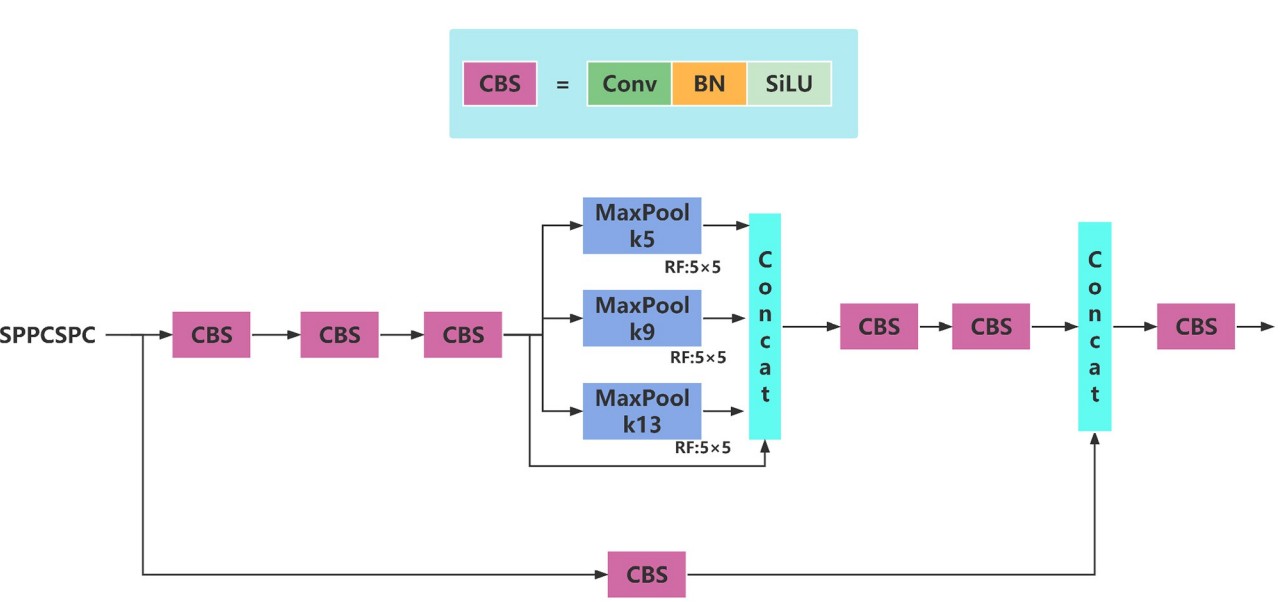

**Fig 5. SPPCSPC structure diagram.**

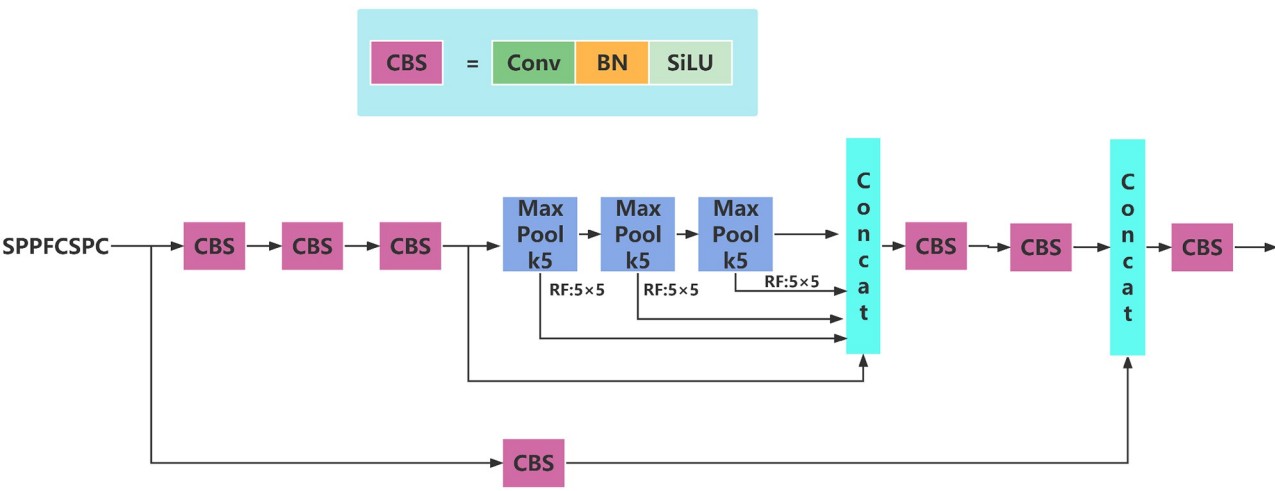

**Fig 6. Structure diagram of SPPFCSPS.**

Cubic convolution interpolation principle:

Assume that the source image A is of size m×n and the target image B scaled K times is of size M×N:

$$K = M/m \tag{1}$$

Each pixel of A is known, but the pixel of B is unknown. To obtain the value of pixel (X,Y) of B, we first need to find the pixel (X,Y) in the source image A corresponding to (x, y);

The 16 nearest pixels from the source image A to the pixel (x, y) are used as the parameters to calculate the pixel value of the target image B(X,Y);

Using the Bi-Cubic basis function to calculate the weights of the 16 pixels, the value of pixel (x, y) in plot B is equal to the weighted superposition of the 16 pixels. According to the scale relation, we can obtain:

$$x/X = m/M = 1/K \tag{2}$$

In the same way, we can also obtain:

$$y/Y = n/N = 1/K \tag{3}$$

B of X,Y corresponds to A:

$$A(x, y) = A(X*(m/M), Y*(n/N)) = A(X/K, Y/K) \tag{4}$$

As shown in the Fig 7, point P is the position of the target image B at (x, y) corresponding to the source image A, and the coordinates of P will have a decimal part. Here, this paper assumes that the coordinates of P are P(x+u, y+v), where x and y are the integer part of the coordinates, and u and v are the decimal part (the distance between the blue point P and the red point in the a11 square in the following Fig 7). Based on this, the position of the nearest 16 pixels can be obtained by using a(i,j) (i,j = 0,1,2,3). In Fig 7, the square is the size of a single pixel, and the red dot is the pixel center.

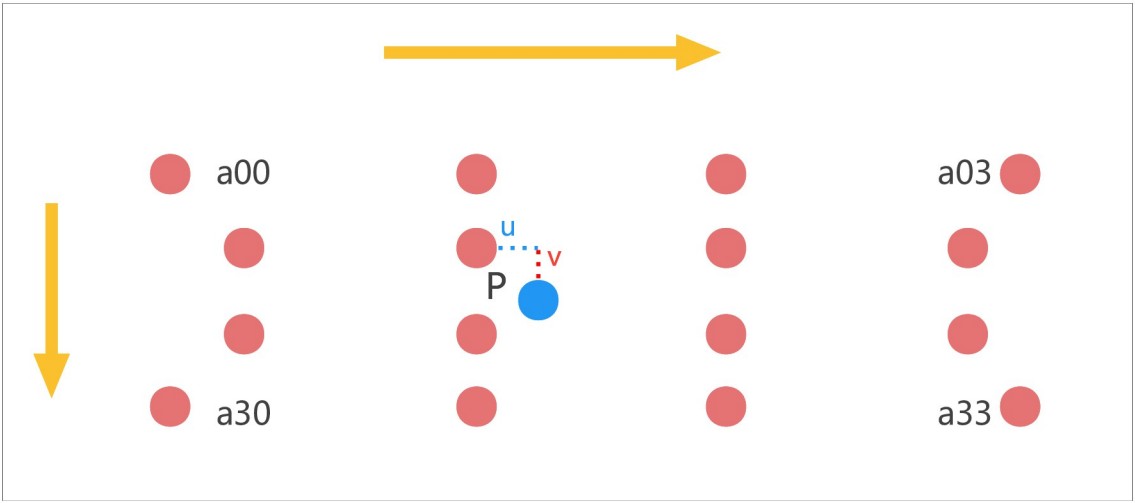

**Fig 7. Pixel location map.**

There are many methods to calculate the weighting coefficient. In this paper, the method based on Bi-Cubic basis function is adopted. The Bi-cubic basis functions are as follows:

$$W(t) \begin{cases} (a+2)|t|^3 - (a+3)|t|^2 + 1 \cdots\cdots for \cdots |t| \leq 1 \\ a|t|^3 - 5a|t|^2 + 8a|t| - 4a \cdots\cdots\cdots for 1 < |t| < 2 \\ 0 \cdots\cdots\cdots\cdots\cdots\cdots\cdots\cdots\cdots otherwise \end{cases} \tag{5}$$

The value of a is -0.5, Calculate the parameter t in the Bi-Cubic function (the distance from the pixel to the point P), so as to obtain the weight W corresponding to the 16 pixels mentioned above. Moreover, the Bi-Cubic basis function is one-dimensional, while the pixel is two-bit, so the row and column of the pixel are calculated separately in this paper.

The pixel value of B(X,Y) is:

$$B(X, Y) = \sum_{I=0}^{3} * \sum_{I=0}^{3} * a_{ij} * W(i) * W(j) \tag{6}$$

Among these three kinds of interpolation, cubic convolution interpolation has better effect and is widely used in many high-quality image transformation. However, in terms of computational speed, cubic convolution interpolation is slower than nearest neighbor interpolation and bilinear interpolation. Nearest neighbor interpolation is used by default in YOLOv5. In this paper, cubic convolution interpolation is used to replace nearest neighbor interpolation in YOLOv5 to obtain better object detection effect. The effect of the three interpolations is shown in Fig 8.

**YOLOv5-plum deep learning network structure.** YOLO(You Only Look Once) is an object detection algorithm proposed in 2016, which is the beginning of One stage algorithm. YOLO is another framework for deep learning object detection speed problem after RCNN and FAST-RCNN. The core idea is to take the whole photo as the input of the network. Regression is performed directly on the position and category of the Bounding Box in the output layer. The biggest advantage of YOLO is that it is extremely fast, which makes it a great advantage in real-time detection tasks. YOLOv5 algorithm [32–34] makes some improvements on

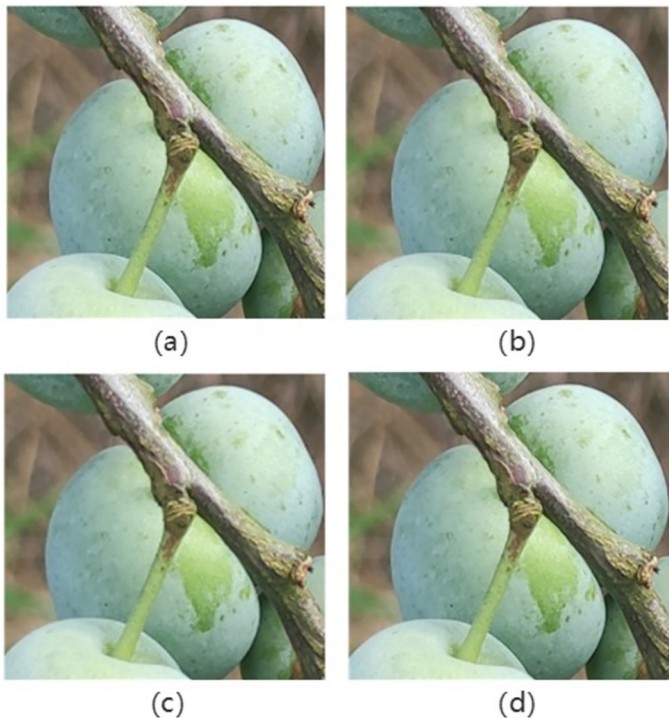

**Fig 8.** Different interpolation effects:(a)is the original image;(b) is the effect of nearest neighbor interpolation;(c) is the effect of bilinear interpolation; and(d) is the effect of cubic convolution interpolation.

the basis of YOLOv4 [35], tasks. YOLOv5 algorithm [32–34] makes some improvements on the basis of YOLOv4 [35], so that its speed and accuracy in target detection are greatly improved. In the model training stage, YOLOv5 is augmented with Mosaic data, and new ideas such as adaptive sight frame calculation and adaptive image scaling are proposed. The excellent ideas of other detection algorithms were integrated in the benchmark network, such as CSP structure and Focus structure. On the Neck network, the FPN+PAN structure was added to YOLOv5. In the Head output layer, the loss function GIOU Loss during training is mainly improved, and the DIOU nms of the prediction box screening is mainly improved. The whole backbone of YOLOv5 is composed of residual convolution. The characteristics of the residual network are easy to optimize, and the accuracy can be improved by increasing the corresponding depth. It alleviates the gradient vanishing problem caused by increasing depth in deep neural networks. Overall, YOLOv5 represents to YOLOv1 YOLOv2, YOLOv3, and the continuous improvement of YOLOv4, YOLOv5 YOLO version of ability was stronger than before, the effect is more outstanding. YOLOv5 contains four kinds of networks, YOLOv5s respectively, YOLOv5l, YOLOv5m, YOLOv5x. In this paper, the fastest YOLOv5s is selected as the original network. Based on the YOLOv5s network, SPPF in the original network is replaced by SPPFCSPC improved in this paper, and the upsampling based on nearest neighbor interpolation is changed to cubic convolution interpolation with better effect. The improved network is shown in Fig 9.

**Multi-scale training.** The size of the input image has a significant impact on the performance of the detection model; in fact, multiscale is one of the most obvious tricks to improve accuracy. In the basic network part, feature maps tens of times smaller than the original image are often generated, so that the feature description of small objects is not easy to be captured

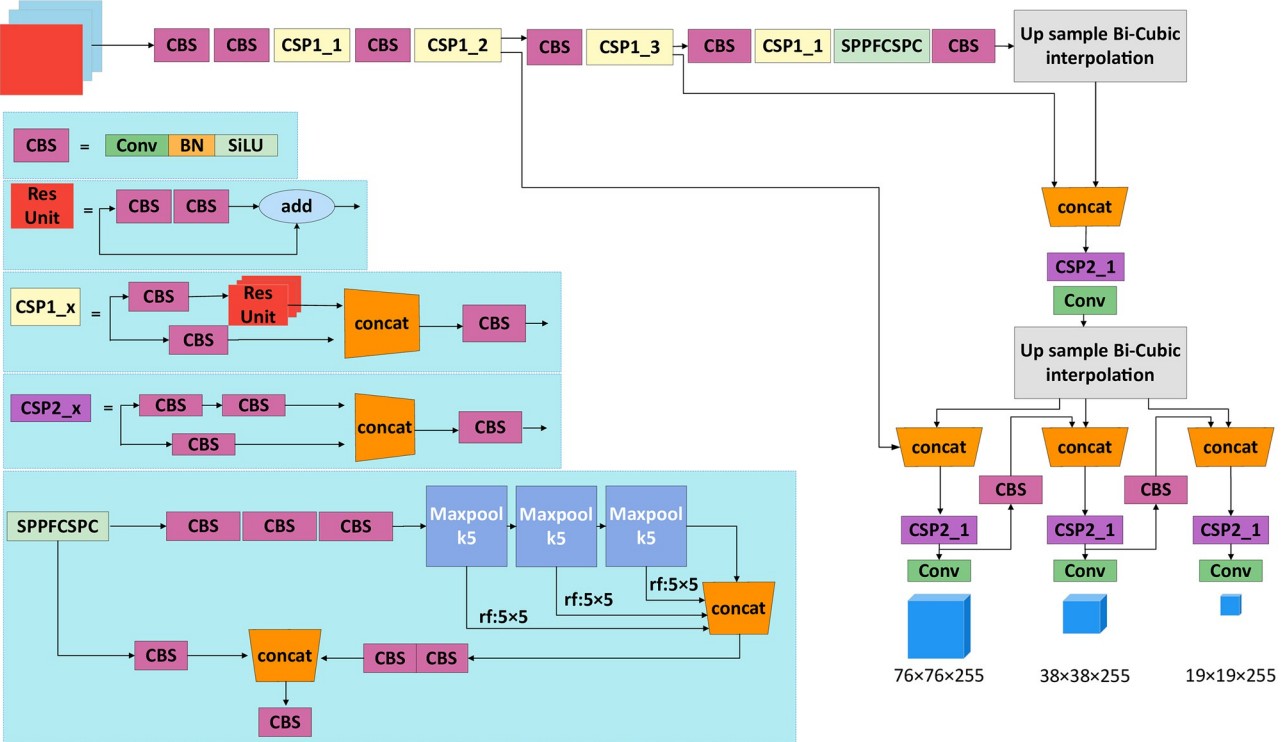

**Fig 9. YOLOv5-plum network structure diagram.**

by the detection network. Training with larger and more sized images can improve the robustness of the detection model to the size of the object to a certain extent. In the multi-scale training of the neural network, several fixed scales are first determined. In each iteration, a scale will be randomly selected and the image will be shrunk or enlarged to that scale size. The images are then imported back into the network for training. In this paper, the input image size is adjusted to 640*640, and then the image is randomly enlarged or reduced by 50% to the network for training, that is, the input image of this paper is trained with three different scales of 320*320, 640*640, 960*960. Three different scale images are shown in Fig 10.

**Evaluation indicators.** In this paper, AP(Average Precision), the most commonly used evaluation metric in object detection, is used as the evaluation metric in this experiment. When AP is mentioned in this paper, we have to mention precision and recall.

Examples are classified into four classes, True positive (TP), based on the combination of their true and predicted class: The prediction result is positive, and the prediction is correct. False negative (FN): The prediction result is negative, but the prediction is incorrect. False positive (FP): The prediction result is positive but the prediction is incorrect, True negative (TN): The prediction result is negative, and the prediction is correct. The four scenarios described above are shown in Table 2.

Precision is a statistic from the perspective of prediction results. It means the proportion of data is truly positive. That is, the percentage of "right search" or precision is calculated as:

$$Precision = \frac{TP}{TP + FP} \tag{7}$$

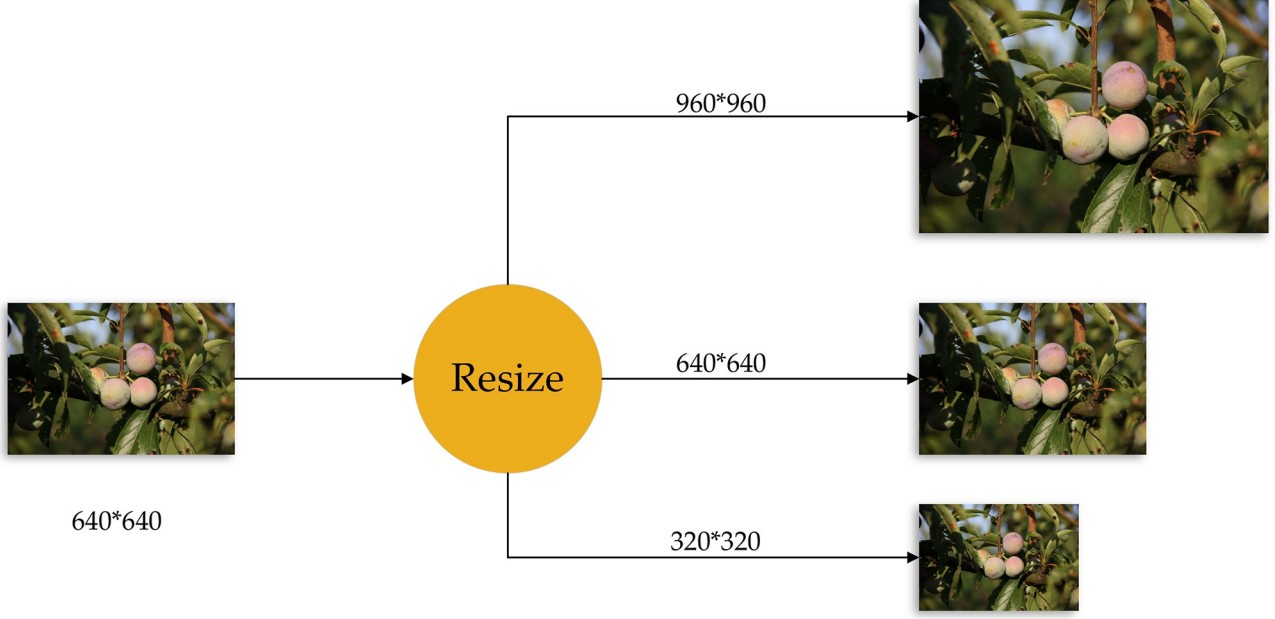

**Fig 10. Illustration of the three scale images.**

The recall is calculated from the real sample set. This means the proportion of the total positive samples recovered by the model. That is, the percentage of "complete searches". Recall is calculated as follows:

$$Recall = \frac{TP}{TP + FN} \tag{8}$$

In the PR curve, P stands for precision and R stands for recall, which represents the relationship between precision and recall. In general, recall is set to the abscissa and precision is set to the ordinate.

AP is the area under the precision-recall (PR) curve, which is also a metric for P-R. The better the classifier, the higher the AP value.

# Results

## Experimental results

After model training, the YOLOv5-plum model has a size of 26.1MB, with an average accuracy of 91.6%. The model was tested on images that were not used for training, and the detection results are shown in Fig 11. Due to factors such as lighting, pests and diseases, and growth

**Table 2. Parameter definitions.**

| Confusion matrix | | Predicted Results | |
|---|---|---|---|
| | | **positive** | **negative** |
| **Actual Results** | positive | **TP** | **FN** |
| | negative | **FP** | **TN** |

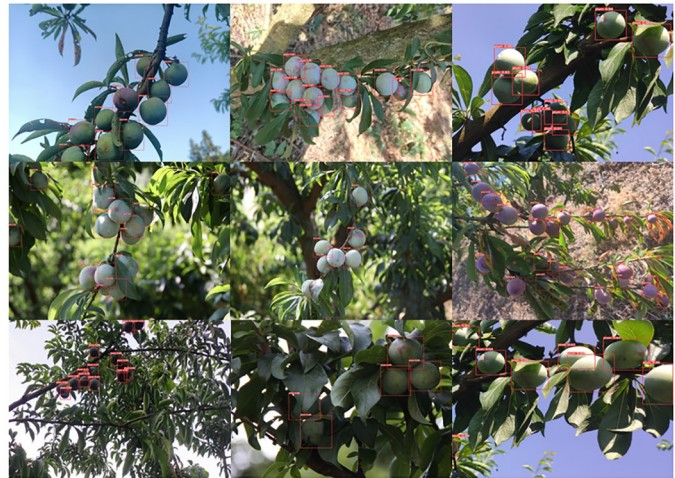

**Fig 11. YOLOv5-plum model detection effect.**

environment, fruit size and density can vary significantly, even within the same period. The images selected for prediction in this study include various shooting distances, angles, and occlusion by leaves. The YOLOv5-plum model demonstrates high accuracy in identifying immature plums in complex natural environments, with a single-image prediction time of only 7.0ms. Therefore, we conclude that the YOLOv5-plum model is capable of accurately and efficiently identifying immature plum fruits and can be combined with other computer vision technologies to facilitate individual pest detection, yield estimation, and other related tasks.

## Comparison of effects of different models

In this study, we compared the performance of the YOLOv5-Plum model with the original YOLOv5 model, and the results are presented in Fig 12. The YOLOv5-Plum model achieved an AP value of 91.60% during detection, indicating better fitting effect, smoother curve, and higher recognition accuracy compared to the traditional YOLOv5 model. Remarkably, despite the accuracy improvement, the model size only increased from 14.4MB to 26.1MB, which still qualifies as an algorithm model with high recognition accuracy and low memory occupation in the field of object detection.

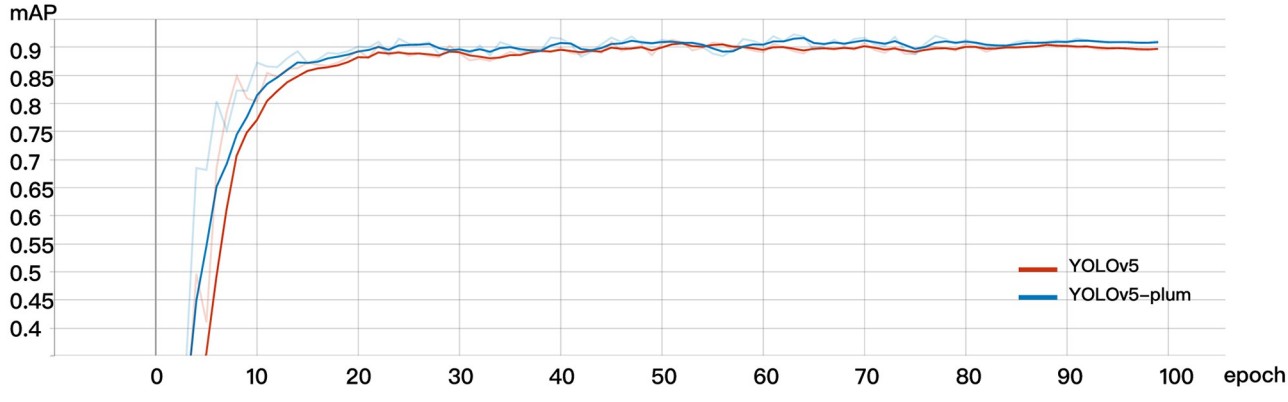

**Fig 12. AP comparison between YOLOv5-plum and YOLOv5.**

**Table 3. Comparison between mainstream detection algorithms.**

| Category | AP | Model size |
|---|---|---|
| YOLOv5 | 0.8980 | **14.4MB** |
| YOLOv7 | 0.9110 | 71.3MB |
| Faster RCNN | 0.9080 | 102MB |
| **YOLOv5-plum** | **0.9160** | 26.1MB |

Then this paper carries out comparative experiments on the current mainstream detection algorithms, and compares the quantitative indicators of YOLOv5-Plum network proposed in this paper with YOLOv5 network, YOLOv7 network and Faster RCNN network, and the comparison results are shown in Table 3.

As shown in the table above, the YOLOv5-Plum model outperforms current mainstream detection algorithms in terms of AP. Compared to the YOLOv5 model, the AP value is increased by 1.8 percentage points, while the model size only increases by 11.7MB. In comparison to the YOLOv7 model and the Faster RCNN model, the AP value is increased by 0.5 percentage points and 0.8 percentage points, respectively, while the model size is 45.2MB and 75.9MB smaller. The YOLOv5-plum model demonstrates higher recognition accuracy and occupies less memory, making it a superior detection method. It can efficiently and accurately complete plum fruit detection tasks and can run on lightweight devices, showcasing its potential for practical applications.

## Ablation study

For the completeness of the experiment and to make sure our model makes sense, We compare the original YOLOv5 model, YOLOv5+Multi model (YOLOv5 plus the model obtained by multi-scale training), and YOLOv5+ Bi model (the upsampling of YOLOv5 is changed to Bi-Cubic interpolation) and YOLOv5+BI+Multi model (original model plus multi-scale training and Bi-cubic interpolation), the four models were trained, the results were compared, and the visualization processing was carried out. Since the operation of the SPPFCSPC module obtained by improving the SPPCSPC structure is only the improvement of speed, its principle is to improve the recognition speed, but not to improve the accuracy, so here we do not train the SPPFCSPC alone, in addition to the original model, All the rest of the training is performed on the basis of SPPFCSPC, and the AP value plot of the training results of the model discussed above is shown in Fig 13.

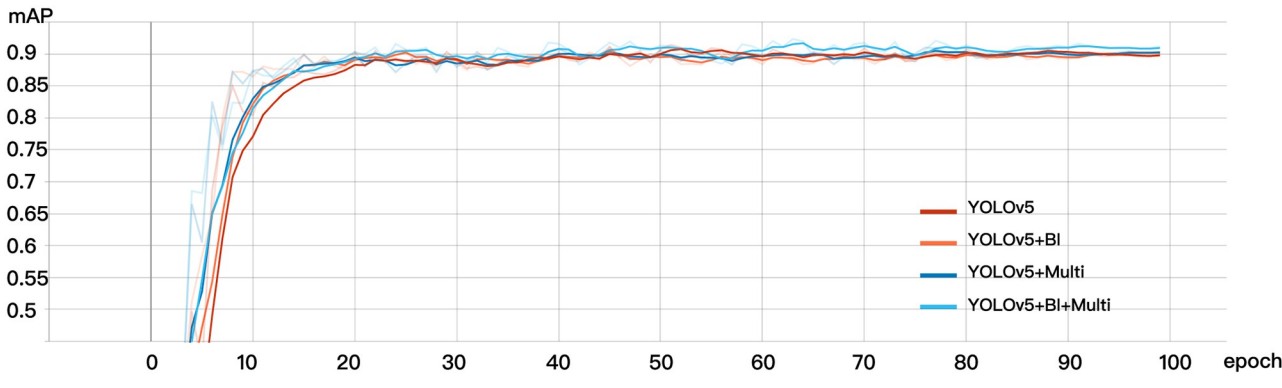

**Fig 13. Comparison of ablation study.**

**Table 4. Table of ablation study results.**

| Model | Multi-scale training | Bi-Cubic interpolation | AP |
|---|---|---|---|
| YOLOv5 | × | × | 0.8980 |
| YOLOv5+Multi | √ | × | 0.9110 |
| YOLOv5+BI | × | √ | 0.9020 |
| YOLOv5+BI+Multi | √ | √ | 0.9160 |

It can be concluded that the combination of multi-scale training and changing the upsampling module to Bi-cubic interpolation can indeed significantly improve the training effect of the model. YOLOv5+BI+Multi model is higher than the other three effects. The AP of YOLOv5+BI+Multi model is approximately 0.5% higher than that of YOLOv5+Multi model, and the AP of YOLOv5+BI+Multi model is approximately 1.4% higher than that of YOLOv5+BI model. It is worth mentioning that the YOLOv5+BI+Multi model is 1.8% higher than that of the original model. The comparison results are shown in Table 4.

### Results of K-fold cross-validation

For small datasets, in order to validate the performance of deep learning models and analyze overfitting, 5-fold cross validation was applied in this study. The results of 5-fold cross validation using the custom YOLO-PLUM model and image preprocessing steps are listed in Table 5. The results show that the custom model performs better, with an average AP of 0.9196. These values are averaged five folds and show very little variance.

## Discussion

### Contribution to plant target detection

The effectiveness of crop models for recognition and detection is highly dependent on the complexity of real-world scenarios. Complex scenes introduce more variables and unknowns, such as light intensity, visibility, leaf occlusion, fruit overlap, camera pixels, and more, all of which can influence the final plant detection model. To improve the adaptability of our model to these complex scenarios, we collected images of immature plums in various environments and carefully annotated and processed the data. The improved algorithm network structure provided excellent feature information that greatly facilitated subsequent experiments. Notably, this study addresses a significant gap in the field of immature plant identification, which has received relatively little research attention to date.

### Contribution to intelligent farming of Crisp red plum

Traditional manual harvesting methods are often inefficient and result in missed fruit, which can be costly. Intelligent picking, on the other hand, can effectively avoid unnecessary harvesting. In this study, we propose an improved YOLOv5 algorithm to develop a YOLOv5-plum model specifically designed for Crisp red plum. Our algorithm enhances detection accuracy by 1.79% and exhibits excellent generalization ability, working well not only on Crisp red plum,

**Table 5. Table of five-fold cross-validation results for the YOLO-PLUM model.**

| Fold Number | 1st-Fold | 2nd-Fold | 3rd-Fold | 4th-Fold | 5th-Fold | Average |
|---|---|---|---|---|---|---|
| AP | 0.9160 | 0.8970 | 0.9430 | 0.9240 | 0.9180 | **0.9196** |

but also on other varieties of plum. In the future, we aim to extend the application of our model to other crops.

## Limits and future work

Admittedly, the study still has limitations. First of all, the location of the data set is only in Sichuan Province, which is relatively simple. In order to make up for this shortcoming, we use different camera equipment to expand the data set, which may not be representative enough. In the future, we will collect data sets of different time periods in multiple regions to do a more comprehensive experiment. Neural network also plays an important role in the prediction of maturity. In the planting process of crops, it is worth studying and digging deeply to infer the ripening period of fruits through the effective recognition of immature fruits. In the future, we plan to predict the ripening period of immature plums through time series, so as to complete the whole intelligent planting chain of plums. The traditional CNN [36] model has a limited ability to process input sequences of variable length, and continuous monitoring based on time series is not feasible. As an alternative method, Recurrent Neural Networks (RNN) allow the retention of time information [37] and can be composed on both spatial and temporal layers. In order to overcome the limitation of the simple RNN model called "vanishing gradient", we further propose the RNN model of long and short sequences [38, 39]. However, LSTM algorithm has the problem of low accuracy and poor effect on long time series. Here, we propose a solution based on LSTF [40] algorithm to predict the maturity period of long time series. In this way, LSTF-RNN will be a promising network, and it is the first time to apply long time series to predict the maturity of plants. Future data may be more informative than we have processed before, which is not only a challenge for us, but also a challenge and even an inspiration for the whole field of smart agriculture.

## Conclusion

In order to enhance the accuracy of identifying immature plums and optimize the model structure of YOLOv5, we aimed to achieve breakthrough improvements that surpass the current state-of-the-art deep learning object detection algorithms. Although YOLOv5 is considered an advanced algorithm with good recognition abilities on most objects, it still faces challenges in terms of slow real-time recognition speed and errors or omissions when detecting small objects. Therefore, we decided to focus on modifying the SPPFCSPC module to tackle these issues.

The SPP module plays a crucial role in avoiding image distortion caused by region cropping and scaling operations, which can negatively impact recognition accuracy. This module effectively resolves the problem of convolutional neural networks extracting repeated features related to graphs, resulting in a significant increase in candidate box generation speed and computational cost savings. By optimizing this module, we can effectively improve recognition speed, leading to better real-time detection performance for immature plums. Our goal is to achieve unprecedented levels of accuracy and performance, surpassing current benchmarks in deep learning object detection.

The upsampling operation in deep learning is essential. Almost all image enlargement uses the interpolation method, that is, on the basis of the original image pixels, a suitable interpolation algorithm is used to insert new elements between the pixels, the main purpose is to make the image can be displayed on higher resolution devices. Nearest neighbor interpolation directly computes the distance between the pixel mapped to the input image coordinate system point and the nearest four points, and assigns the pixel value of the nearest pixel to the input image coordinate system point. The nearest neighbor interpolation method is indeed fast, but

the new image locally destroys the gradient relationship of the original image. Bilinear interpolation first uses linear interpolation in one direction, and then uses linear interpolation in the other direction. Although the steps are linear in the sampling value and position, the overall interpolation is not linear, but quadratic in the sampling position, which makes the final bilinear interpolation in the processing of image data richer than the results of single linear interpolation. This is one of the reasons why bilinear interpolation is chosen for model improvement in this paper.

So as to improve the robustness of the model, we consider that the size of the input image has different effects on the performance of different models, and multi-scale training can train by insetting larger and more size images. This method is one of the most obvious ways to improve the accuracy, so we add the model to multi-scale training, and finally obtain the YOLOv5-plum model. The above results demonstrate the feasibility of making these improvements in YOLOv5. In addition, YOLOv5-plum still has room for improvement in lightweight. This study not only provides an efficient and accurate plum detection model for growers, but also verifies the feasibility and great development potential of computer vision in the field of agricultural products detection. In addition, this kind of model can be applied to other immature crops or fruits, which is helpful to establish a complete and efficient crop recognition system.

As this study focuses on the detection of one specific type of plum, future research could explore the extension of this method to other plum varieties with different colors and appearances. Additionally, for improved image acquisition, sensors with higher quality, such as hyperspectral cameras, could be utilized. Further experiments could also be conducted, such as hyperspectral prediction of plum yield, to expand the potential applications of this technology.

## Acknowledgments

Thanks to all partners in AI Studio for their support.

## Author Contributions

**Conceptualization:** Yupeng Niu.

**Data curation:** Yupeng Niu, Jiong Mu.

**Formal analysis:** Ming Lu.

**Funding acquisition:** Jiong Mu.

**Investigation:** Ming Lu.

**Methodology:** Yupeng Niu, Qianqian Wu.

**Project administration:** Qianqian Wu.

**Resources:** Yupeng Niu, Qianqian Wu, Jiong Mu.

**Software:** Yupeng Niu, Xinyun Liang.

**Supervision:** Xinyun Liang, Jiong Mu.

**Validation:** Yupeng Niu, Xinyun Liang.

**Visualization:** Qianqian Wu.

**Writing – original draft:** Yupeng Niu, Ming Lu, Jiong Mu.

**Writing – review & editing:** Qianqian Wu, Jiong Mu.

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
