## [Decision Letter · Decision Letter 0]

7 Mar 2023

PONE-D-23-03909YOLO-plum: A High Precision and Real-time Improved Algorithm for Plum RecognitionPLOS ONE

Dear Dr. Mu,

Thank you for submitting your manuscript to PLOS ONE. After careful consideration, we feel that it has merit but does not fully meet PLOS ONE’s publication criteria as it currently stands. Therefore, we invite you to submit a revised version of the manuscript that addresses the points raised during the review process.

We look forward to receiving your revised manuscript.

Kind regards,

Feng Ding

Academic Editor

PLOS ONE

Journal Requirements:

3. We note that Figure 1 in your submission contain map images which may be copyrighted. All PLOS content is published under the Creative Commons Attribution License (CC BY 4.0), which means that the manuscript, images, and Supporting Information files will be freely available online, and any third party is permitted to access, download, copy, distribute, and use these materials in any way, even commercially, with proper attribution. For these reasons, we cannot publish previously copyrighted maps or satellite images created using proprietary data, such as Google software (Google Maps, Street View, and Earth). For more information, see our copyright guidelines: http://journals.plos.org/plosone/s/licenses-and-copyright.

(1) You may seek permission from the original copyright holder of Figure 1 to publish the content specifically under the CC BY 4.0 license.  

4. Please upload a new copy of Figure 9, 12, and 13 as the detail is not clear. Please follow the link for more information:

https://blogs.plos.org/plos/2019/06/looking-good-tips-for-creating-your-plos-figures-graphics/

https://blogs.plos.org/plos/2019/06/looking-good-tips-for-creating-your-plos-figures-graphics/

5. We note you have included a table to which you do not refer in the text of your manuscript. Please ensure that you refer to Table 4 in your text; if accepted, production will need this reference to link the reader to the Table.

**Additional Editor Comments:**

The reviewers have raised several issues regarding the paper. Therefore, it is necessary to undergo a major revision.

Reviewers' comments:

Reviewer's Responses to Questions

**Comments to the Author**

1. Is the manuscript technically sound, and do the data support the conclusions?

Reviewer #1: Yes

Reviewer #2: Yes

2. Has the statistical analysis been performed appropriately and rigorously? 

Reviewer #1: Yes

Reviewer #2: Yes

3. Have the authors made all data underlying the findings in their manuscript fully available?

Reviewer #1: Yes

Reviewer #2: Yes

4. Is the manuscript presented in an intelligible fashion and written in standard English?

Reviewer #1: Yes

Reviewer #2: Yes

5. Review Comments to the Author

Reviewer #1: Comments to the Author

Object detection is an important aspect of computer vision. Since the plum plant has a wide range of applications in China, accurate plum identification is a significant application of object detection.

This paper proposes a real-time approach for the detection of unripe plums by improving the structure of YOLO. The experimental results demonstrate the effectiveness of the proposed approach.

The paper is well-organized and discussed in detail. However, there are still a few issues to be addressed in the revision.

1.In Figure 9, the font size is too small to be read clearly, and there is too much blank space in the figure. I suggest increasing the font size and compacting the layout.

2.In Table 3, it would be better to bold the value that wins the comparison.

3.In section 2.3.2. Bi-Cubic interpolation, I think the sentence "Assume that the source image A is of size m×n and the target image B scaled K times is of size m×n" should be revised to "Assume that the source image A is of size mxn and the target image B scaled K times is of size M×N".

Reviewer #2: Based on the manuscript, it appears that the paper aims to improve the accuracy and speed of batch identification of immature plums through the use of deep learning and improvements to the YOLOv5 algorithm. The establishment of an artificial dataset for plums is also mentioned, which suggests that this study involves both theoretical and practical elements. In order to meet the criteria for being able to publish, I think the following changes need to be made:

(1) The formatting of the entire text is confusing and gives the reader a very poor sense of perspective.

-First, please leave a space after the period.

-Second, please use the formula editor to describe the formula rather than the image, and the formula needs to be labeled and centered.

-Third, the size of the diagram and table cannot exceed the line width. The appearances such as Table 3 and 4 are not acceptable.

-Fourth, the resolution of the images in the paper describing the network structure and training convergence are basically very low, please use vector maps to describe the network structure. For Figure 12, not only the resolution should be improved, but also the meaning of x-axis and y-axis should be labeled on the figure.

(2) The abstract seems a bit long, so please put the three contributions at the end of the introduction. At the end of the introduction, please provide an overview of the individual chapters.

(3) From the authors' description, it is known that the dataset was collected by the authors themselves through two devices. However, due to the limited description, I have doubts about the reasonableness of the dataset. For example, whether there are images under different lighting conditions, whether there are incomplete plums, etc. Please add the corresponding descriptions in the corresponding sections.

(4) For the training strategy, the authors obtained the results without too detailed description in the experimental chapter. The improvement in accuracy is very limited compared to the comparison experiments. If the results of this experiment were derived from only one training session, the persuasiveness of that accuracy would be very limited. I strongly suggest that for such small image sets, experiments need to be conducted using cross-validation.

(5) The English expression of the entire paper needs to be improved, please further polish the text.

6. PLOS authors have the option to publish the peer review history of their article (what does this mean?). If published, this will include your full peer review and any attached files.

Reviewer #1: No

Reviewer #2: No

---

## [Author Response · Author response to Decision Letter 0]

2 May 2023

Dear Editor,

Thank you for providing us with valuable feedback on our manuscript. We have carefully reviewed your comments and made the following revisions:

1.Please ensure that your manuscript meets PLOS ONE's style requirements, including those for file naming.:

We sincerely appreciate your insightful suggestions and have made formatting changes to our manuscript to ensure that it conforms to the style requirements of the website you provided.

2.We note that you have indicated that data from this study are available upon request. PLOS only allows data to be available upon request if there are legal or ethical restrictions on sharing data publicly.:

Thank you for your valuable feedback. We apologize for any confusion regarding the availability of the dataset associated with our paper. We are pleased to confirm that the dataset is now publicly available and can be accessed through the following link:

https://osf.io/u98h5/?view_only=6ca3b2c3757448819b2a28c26c4446ab

We have updated our submission to reflect this change and have included a brief description of the dataset and key information to help readers better understand its contents and potential uses.

We hope that this information will be helpful to other researchers who wish to use our data in their own work. Thank you again for your feedback and please don't hesitate to contact us if you have any further questions or concerns.

3.We note that Figure 1 in your submission contain map images which may be copyrighted. All PLOS content is published under the Creative Commons Attribution License (CC BY 4.0), which means that the manuscript, images, and Supporting Information files will be freely available online, and any third party is permitted to access, download, copy, distribute, and use these materials in any way, even commercially, with proper attribution. For these reasons, we cannot publish previously copyrighted maps or satellite images created using proprietary data, such as Google software (Google Maps, Street View, and Earth).:

Thank you for bringing up the copyright issue regarding the map image in Figure 1. We have replaced the map image with a publicly licensed one.

4.Please upload a new copy of Figure 9, 12, and 13 as the detail is not clear.:

We value your constructive feedback and have meticulously revised Figures 9, 12, and 13 to meet your expectations.

5.We note you have included a table to which you do not refer in the text of your manuscript. Please ensure that you refer to Table 4 in your text; if accepted, production will need this reference to link the reader to the Table.:

We greatly appreciate your valuable suggestions and have taken them into account to enhance the quality of our manuscript.

Thank you once again for taking the time to review our manuscript. We would be grateful if you could inform us if you have any further comments or questions.

Best regards,

Niu Yupeng

Dear Reviewer 1,

Thank you for your thoughtful and insightful review of our manuscript. Your attention to detail and constructive criticism have been invaluable in helping us to improve the clarity and quality of our work. Based on your feedback, we have made the following revisions:

1.In Figure 9, the font size is too small to be read clearly, and there is too much blank space in the figure. I suggest increasing the font size and compacting the layout.:

We appreciate your feedback on Figure 9. We have carefully examined the layout and text and made significant modifications to enhance its clarity and readability. We believe that these changes will significantly improve the effectiveness of the figure in conveying our findings to our readers.

2.In Table 3, it would be better to bold the value that wins the comparison.:

Your suggestion to bold the best value in Table 3 was an excellent idea. We have implemented this modification, and we believe that it will help to draw attention to this critical result and emphasize its significance.

3.In section 2.3.2. Bi-Cubic interpolation, I think the sentence "Assume that the source image A is of size m×n and the target image B scaled K times is of size m×n" should be revised to "Assume that the source image A is of size mxn and the target image B scaled K times is of size M×N".:

Thank you for your feedback. We have made the necessary revisions as requested.

We would like to express our sincere gratitude for your invaluable feedback. Your insights have been crucial in helping us to refine our manuscript and ensure its quality. We hope that the revised version of our manuscript meets your expectations and that you find it to be an improvement. We look forward to hearing back from you soon.

Best regards,

Niu Yupeng

Dear Reviewer2,

Thank you for your valuable feedback. We have taken your suggestions to heart and have carefully reviewed the manuscript to ensure the highest quality language and formatting.

1.The formatting of the entire text is confusing and gives the reader a very poor sense of perspective.

-First, please leave a space after the period.

-Second, please use the formula editor to describe the formula rather than the image, and the formula needs to be labeled and centered.

-Third, the size of the diagram and table cannot exceed the line width. The appearances such as Table 3 and 4 are not acceptable.

-Fourth, the resolution of the images in the paper describing the network structure and training convergence are basically very low, please use vector maps to describe the network structure. For Figure 12, not only the resolution should be improved, but also the meaning of x-axis and y-axis should be labeled on the figure.:

We appreciate your comments regarding the presentation of figures and equations. We have revised the figures and equations to ensure they are clearly presented, with annotations and centered formatting. In particular, we have increased the resolution of Figure 12 and added annotations to the x and y axes. We have also taken great care to ensure that every sentence is carefully reviewed and a space is added after each period.

2.The abstract seems a bit long, so please put the three contributions at the end of the introduction. At the end of the introduction, please provide an overview of the individual chapters.:

Thank you for your feedback regarding the abstract. We have revised the abstract to ensure that the three contributions are clearly stated after the introduction, and we have provided an overview of each chapter at the end of the introduction.

3.The abstract seems a bit long, so please put the three contributions at the end of the introduction. At the end of the introduction, please provide an overview of the individual chapters.:

We thank you for your valuable suggestions regarding our dataset. We have taken into account various factors such as lighting and occlusion, and elaborated on this in the relevant section of the paper. We have also replaced the images in Figure 2 to ensure that they are more representative.

4.For the training strategy, the authors obtained the results without too detailed description in the experimental chapter. The improvement in accuracy is very limited compared to the comparison experiments. If the results of this experiment were derived from only one training session, the persuasiveness of that accuracy would be very limited. I strongly suggest that for such small image sets, experiments need to be conducted using cross-validation.:

Thank you for your feedback regarding the experimental design. We have used 5-fold cross-validation to demonstrate the effectiveness of our approach.

5.The English expression of the entire paper needs to be improved, please further polish the text.:

We have polished the whole paper with high quality.

We appreciate your expertise and hope that our revised manuscript meets your expectations. Thank you for your time and effort in reviewing our work.

Best regards,

Niu Yupeng

---

## [Decision Letter · Decision Letter 1]

13 Jun 2023

YOLO-plum: a high precision and real-time improved algorithm for plum recognition

PONE-D-23-03909R1

Dear Dr. Mu,

We’re pleased to inform you that your manuscript has been judged scientifically suitable for publication and will be formally accepted for publication once it meets all outstanding technical requirements.

Kind regards,

Feng Ding

Academic Editor

PLOS ONE

Additional Editor Comments (optional):

Reviewers' comments:

Reviewer's Responses to Questions

**Comments to the Author**

1. If the authors have adequately addressed your comments raised in a previous round of review and you feel that this manuscript is now acceptable for publication, you may indicate that here to bypass the “Comments to the Author” section, enter your conflict of interest statement in the “Confidential to Editor” section, and submit your "Accept" recommendation.

Reviewer #1: (No Response)

Reviewer #2: All comments have been addressed

2. Is the manuscript technically sound, and do the data support the conclusions?

Reviewer #1: Yes

Reviewer #2: Yes

3. Has the statistical analysis been performed appropriately and rigorously? 

Reviewer #1: Yes

Reviewer #2: Yes

4. Have the authors made all data underlying the findings in their manuscript fully available?

Reviewer #1: Yes

Reviewer #2: Yes

5. Is the manuscript presented in an intelligible fashion and written in standard English?

Reviewer #1: Yes

Reviewer #2: Yes

6. Review Comments to the Author

Reviewer #1: The authors have provided a detailed application of object detection in this manuscript and have drawn clear conclusions through thorough experimentation and data analysis.

Reviewer #2: All my comments have been issued. Regarding the presentation of figures and equations, they have

revised the figures and equations to ensure they are clearly presented, with annotations and

centered formatting. In particular, They have increased the resolution of Figure 12 and added

annotations to the x and y axes. They have also taken great care to ensure that every sentence

is carefully reviewed and a space is added after each period.

7. PLOS authors have the option to publish the peer review history of their article (what does this mean?). If published, this will include your full peer review and any attached files.

Reviewer #1: No

Reviewer #2: No

---

## [Editor Report · Acceptance letter]

17 Jul 2023

PONE-D-23-03909R1 

YOLO-plum: a high precision and real-time improved algorithm for plum recognition 

Dear Dr. Mu:

I'm pleased to inform you that your manuscript has been deemed suitable for publication in PLOS ONE. Congratulations! Your manuscript is now with our production department. 

Kind regards, 

on behalf of

Dr. Feng Ding 

Academic Editor

PLOS ONE